# Revisiting the Morphology and Classification of the Paracingulate Gyrus with Commentaries on Ambiguous Cases

**DOI:** 10.3390/brainsci11070872

**Published:** 2021-06-29

**Authors:** Grzegorz Wysiadecki, Agata Mazurek, Jerzy Walocha, Agata Majos, R. Shane Tubbs, Joe Iwanaga, Andrzej Żytkowski, Maciej Radek

**Affiliations:** 1Department of Normal and Clinical Anatomy, Chair of Anatomy and Histology, Medical University of Lodz, 90-752 Łódź, Poland; 2Department of Anatomy, Jagiellonian University Medical College, 33-332 Kraków, Poland; agt.mazurek@student.uj.edu.pl (A.M.); jwalocha@cm-uj.krakow.pl (J.W.); 3Department of Radiological and Isotopic Diagnosis and Therapy, Medical University of Lodz, 92-213 Łódź, Poland; agata.majos@umed.lodz.pl; 4Tulane Center for Clinical Neurosciences, Department of Neurosurgery, Tulane University School of Medicine, New Orleans, LA 70112, USA; shane.tubbs@icloud.com (R.S.T.); iwanagajoeca@gmail.com (J.I.); 5Department of Neurosurgery and Ochsner Neuroscience Institute, Ochsner Health System, New Orleans, LA 70433, USA; 6Tulane Center for Clinical Neurosciences, Department of Neurology, Tulane University School of Medicine, New Orleans, LA 70112, USA; 7Department of Anatomical Sciences, St. George’s University, Grenada FZ 818, Grenada; 8Department of Surgery, Tulane University School of Medicine, New Orleans, LA 70112, USA; 9Department of Anatomy, Kurume University School of Medicine, 67 Asahi-machi, Kurume, Fukuoka 830-0011, Japan; 10Department of Polish Dialectology and Logopedics, Faculty of Philology, University of Lodz, 90-236 Łódź, Poland; azytkowski@protonmail.com; 11Norbert Barlicki Memorial Teaching Hospital No. 1 of the Medical University of Lodz, 90-153 Łódź, Poland; 12Department of Neurosurgery, Spine and Peripheral Nerve Surgery, Medical University of Lodz, University Hospital WAM-CSW, 90-549 Łódź, Poland; maciej.radek@umed.lodz.pl

**Keywords:** anatomical variations, cerebral cortex, human brain, neuroscience, neuroanatomy, paracingulate gyrus

## Abstract

The anterior cingulate cortex is considered to play a crucial role in cognitive and affective regulation. However, this area shows a high degree of morphological interindividual variability and asymmetry. It is especially true regarding the paracingulate sulcus and paracingulate gyrus (PCG). Since the reports described in the literature are mainly based on imaging techniques, the goal of this study was to verify the classification of the PCG based on anatomical material. Special attention was given to ambiguous cases. The PCG was absent in 26.4% of specimens. The gyrus was classified as present in 28.3% of cases. The prominent type of the PCG was observed in 37.7% of the total. Occasionally, the gyrus was well-developed and roughly only a few millimeters were missing for classifying the gyrus as prominent, as it ended slightly anterior the level of the VAC. The remaining four cases involved two inconclusive types. We observed that the callosomarginal artery ran within the cingulate sulcus and provided branches that crossed the PCG. Based on Klingler’s dissection technique, we observed a close relationship of the PCG with the superior longitudinal fascicle. The awareness of the anatomical variability observed within the brain cortex is an essential starting point for in-depth research.

## 1. Introduction

Brain mapping techniques constantly evolve and rely on developing and refining methods for acquiring, representing, analyzing, visualizing, and interpreting the radiological image. For decades, a significant amount of clinical and experimental data have been accumulated stating that specific symptoms can occur due to damage to some regions of the brain, but the location of these areas was often only an approximation [1]. Admittedly, it has long been clear that the presumptions regarding narrow localization of specific brain functions is a significant generalization. The phenomena such as speaking, understanding speech, writing, reading, or seeing are not simple, elementary processes [1,2,3,4]. On the contrary, these processes are complex activities that comprise minor phenomena that involve various brain areas and can be referred to as brain processing patterns [2]. In this context, current functional and structural neuroimaging available can support medical, psychological, and brain–mind relationship studies [3,4,5]. Developing modern diagnostic techniques, including functional magnetic resonance imaging and tractography, has opened up new possibilities in understanding the anatomy of the brain and the relationship between different areas. However, as Stufflebeam and Rosen [3] stressed, a complete understanding of cognition only comes through integrating multimodal structural and functional imaging modalities. A thorough knowledge of anatomy is the basis for the correct interpretation of imaging results.

Structural and functional imaging of the brain faces several methodological difficulties. One of the main problems is that the anatomy of the human body, including the human brain’s structure, cannot reliably be reflected in generalized and simplified diagrams or models [6]. Anatomical variability and subtle differences in the morphological appearance of certain structures are commonly observed between individuals. When making a classification, a very fine line between specific anatomical variants can occasionally be observed [6]. An example is the classification of the paracingulate gyrus.

The anterior cingulate cortex is considered to play a crucial role in cognitive and affective regulation. However, as Wei et al. [7] stressed, this area shows a high degree of morphological interindividual variability and asymmetry. It is especially true regarding the paracingulate sulcus (PCS) and paracingulate gyrus (PCG). Three types of PCG, i.e., prominent, present, or absent, were proposed by Yücel et al. [8]; such variants of the PCG and sulcus are typically distinguished in reports in the literature [7,8,9,10,11,12]. Since the reports described in the literature are mainly based on imaging techniques, the goal of this study was to verify the classification of the PCG based on anatomical material. Special attention was given to ambiguous cases. Klingler’s method of brain dissection and coronal brain sectioning was also applied to the research.

## 2. Materials and Methods

The study was conducted according to the guidelines of the Declaration of Helsinki and approved by the Bioethics Committee of the Medical University of Lodz (protocol code: RNN/515/14KB). Fifty-three isolated adult cerebral hemispheres (26 right and 27 left) fixed in 10% formalin were examined. The spatial orientation was referred to as anterior commissure-posterior commissure baseline (i.e., the Talairach–Tournoux baseline; AC-PC line) and vertically erected perpendicular to the AC-PC baseline at the anterior commissure (VAC line) [13,14]. The anatomical classification used in this study was adapted from Yücel et al. [8]. To classify the PCG as “present,” a PCS localized dorsal and parallel to the cingulate sulcus was mandatory and a continuous length of the sulcus should be at least 20 mm. According to the classification proposed by Yücel et al. [8], the PCG was classified as “prominent” when the PCS was longer than 40 mm. Since this approach is not sharp, in our study, we used the VAC line as a landmark for distinguishing “present” (ending forward from this line) from “prominent” (crossing this line). Image J image processing program (developed at the National Institutes of Health and the Laboratory for Optical and Computational Instrumentation) was used to measure the PCS length in ambiguous cases. Klingler’s method of brain dissection and coronal brain sectioning were also used on three specimens, according to earlier descriptions [15,16,17,18].

## 3. Results

The PCG was absent in 14 specimens (26.4%; in seven right hemispheres = 26.9% and in seven left hemispheres = 25.9%; see Figure 1). In 15 out of all examined specimens, the gyrus was present (28.3% of the total; seven right = 26.9% of right hemispheres and eight left = 29.6% of left hemispheres; Figure 2). On the examined sample, the prominent type of the PCG (Figure 3 and Figure 4) was observed in 20 out of 53 specimens (37.7% of the total; ten right = 38.5% of right hemispheres and ten left = 37% of left hemispheres). Occasionally, the gyrus was well-developed and only a few millimeters were missing for classifying the gyrus as prominent, as it ended slightly anterior to the level of the VAC line (Figure 5). The remaining four cases (two right and two left hemispheres) involved inconclusive (irregular or difficult to classify) types, two on the right and two on the left side. Among those difficult to classify cases, two specimens (one right and one left) were found to have an underdeveloped morphology of the cingulate gyrus (Figure 6). This type could be classified as being absent (PCS < 20 mm). From an anatomical point of view, i.e., based on dissection conducted according to Klingler’s technique, it can also be classified as an “underdeveloped type.” The PCS was interrupted in 9 out of 20 gyri of prominent type, which also caused ambiguity during evaluation (Figure 7). We observed that the callosomarginal artery ran within the cingulate sulcus and provided branches that crossed the PCG in specimens with preserved arteries (Figure 3A and Figure 8). However, the callosomarginal and pericallosal arteries were highly variable regarding their branching pattern and diameters. Based on Klingler’s dissection technique, we observed a close relationship of the PCG (when present) with the superior longitudinal fascicle (Figure 4B).

## 4. Discussion

The PCS has been considered a human-specific anatomical structure involved with higher-order cognitive processing, but some studies have identified the PCS in some primates’ brains [5]. According to previous neuroanatomical reports, the morphology of the paracingulate area might be classified depending on morphometry of the PCS with the symmetric pattern or left-/rightward asymmetry as follows: “prominent,” “present,” or “absent” [7,8,9,10]. Yücel et al. 2001 [8], Fornito et al. [10], and Amiez et al. [5] reported that there is a higher prevalence of “prominent” or “present” PCG types in the left hemisphere, while the leftward pattern might be more frequent in males compared to females, who were more likely to have a symmetrical pattern [8]. However, the incidence of PCG reported in different studies varies. ten Donkleaar et al. [19] stress that frequently, “a series of furrows delineates the paracingulate sulcus (*sulcus paracinguli*), which separates the medial division of the superior frontal gyrus from the paracingulate gyrus (*gyrus paracinguli*)”; those authors determine the mutually parallel course of the cingulate and paracingulate gyri as “a double-parallel pattern, where the paracingulate sulcus surrounds the cingulate sulcus.” Ono et al. [20] reported the frequency of the PCG occurrence to be 24%. Contrary to those results, Yücel et al. [8] found the prominent, present, and absent PCG accordingly in 49%, 25%, and 26% of the left hemispheres; the same types of the PCG were observed in 28%, 34%, and 38% of the right hemispheres, respectively. In our study, the frequencies of the PCG variants were similar in both hemispheres. However, when assessing these results, it should be remembered that the study was conducted on an atypical sample (isolated cerebral hemispheres). Such a sample cannot be treated as representative of the population in terms of statistical criteria. For statistical purposes, the whole brains should be examined (see study limitation). However, our research was aimed at anatomical verification of PCG classification and not for statistical assessment of differences of PCG morphology between cerebral hemispheres. We proposed complementary criteria of confirmation of the presence of the PCG, such as close relationships of the PCG with the superior longitudinal fascicle.

Cingulate sulcus (CS) variability is described in reports in the literature besides the PCS basing on the classification of Paus et al. [12], who distinguished the types of CS into “continuous” and “interrupted” (distinct interruption >10 mm). Those authors evaluated the presence and location of such morphological CS features as the continuity of the CS, the presence of vertically oriented branches of the CS, the presence of the PCS, and the presence of the intralimbic sulcus; They also depicted the presence of several types of vertically oriented CS “branches” with length ≥10 mm [12]. Analysis of the PCS and CS anatomy with measurements of gray matter volume and gray matter thickness of the medial prefrontal cortex allows for assessing subtle differences of cortical folding between healthy individuals and those with neuropsychiatric disorders. For instance, Provost et al. [21] reported significant asymmetry in the brains between healthy volunteers, whose PCS were more common and more marked in left hemispheres, and men with early onset schizophrenia, whose PCS were as frequent in the right as in the left hemispheres.

The superior longitudinal fascicle (SLF) is a distinct fiber tract running within the cingulate or paracingulate gyrus and connecting the anterior cingulate cortex, the medial aspect of the superior frontal gyrus, the pre-supplementary motor area (pre-SMA), the SMA proper, the paracentral lobule, and the precuneus [22,23]. Such connections support findings revealing increased activity in this area during speech, eye movements [24], cognitive-control-related processes, pain, negative affect, and planning movements [25,26,27]. Yagmurlu et al. [22] examined segmentation of the SLF and found that SLF I had a close anatomical relationship with the cingulum at the medial side of the hemisphere; however, it courses parallel to the cingulate sulcus, and it does not reach the cingulum. This is consistent with later findings of Komaitis et al. [23], who suggested that the white matter within the PCG (if present) is represented mainly by subcomponent Ia of the superior longitudinal fascicle [23]; based on those authors’ findings, the SLF-Ia was always seen to course within the paracingulate gyrus, when this gyrus was present, or within the anterior part of the cingulate gyrus and deep to the cingulate sulcus in any other instance. As Komaitis et al. (p. 1274, [23]) summarizing their findings stress that “Our results reveal that the SLF-I represents a distinct white matter tract, always traveling under the superficial U-fibers of the cingulate or paracingulate gyrus and connecting the cortical hubs of the medial cerebral hemisphere, that is, the anterior cingulate cortex (BA32), the medial frontal gyrus (medial aspect of BA6, -8, -9), the paracentral lobule (BA1, -2, -3, -4), and the superior parietal lobule and precuneus (BA5 and -7).” Brodmann’s area 32 differs in its cytoarchitectonic structure compared to the adjacent BA6 and BA24 areas [28]. Measurements of cortical thickness within this area might help identify patients with major depressive disorder (MDD) who tend to have decreased thickness in BA32 and BA24 in the right hemisphere compared to healthy controls, and reduction of gray matter within BA32 might be correlated to depression. Loss of gray matter in BA32 and BA24 in MDD patients is considered to correlate with concentrations of C-reactive protein (CRP) and neurotoxic metabolites of kynureine [29]. Yagmurlu et al. [22] also suggested that the primary subcortical connection of the default mode network can take place through the SLF I. This long association bundle is connected to the precuneus and potentially to the anterior cingulate region. The precuneus plays an important role in self-consciousness concerning the default mode network, while lesions within the anterior cingulate cortex may be entirely or partly related to the default mode network. At this point, the question arises whether different types of PCG formation can be associated with different anatomical variants of the SLF, which should be verified in further in-depth studies using tractography and Klingler’s technique of brain dissection.

Cortical thickness within the PCG increases from childhood to adulthood [30]. The presence of the PCG was associated with significantly larger paracingulate cortex volume with significantly decreased anterior cingulate cortex volume within the same hemisphere [31]. Leftward asymmetry of the PCS might be correlated with better spatial working memory both in patients with schizophrenia and healthy controls [31]. Gray matter volume within the PCG might be increased in patients with schizophrenia and hallucinations. Schizophrenia patients with a history of hallucinations have been found to have a significantly reduced length of the PCS in the left hemisphere in comparison with patients with schizophrenia not complaining of hallucinations and also a significantly smaller local gyrification index of the medial prefrontal cortex adjacent to the PCS with larger gray matter volume in this region, suggesting altered morphology of the medial prefrontal (and paracingulate) cortex associated with this psychiatric disorder. Both groups of patients were found to have a significantly reduced length of the PCS in both hemispheres compared to controls [32]. According to Garrison et al. [33], hallucinations can be associated with specific brain morphology when comparing patients with schizophrenia who have experienced hallucinations to patients who have not; according to these authors, a 1 cm reduction in PSC length “increased the likelihood of hallucinations by 19.9%, regardless of the sensory modality in which they were experienced. The findings suggest a specific morphological basis for a pervasive feature of typical and atypical human experience.” Individuals with a higher genetic risk of psychosis were found to have a decreased occurrence of the PCS in comparison with healthy controls and slightly higher compared with patients with schizophrenia [34]. Koo et al. [35] found that a significant difference in the PCS asymmetry and loss of gray matter within the cingulate gyrus is observed in patients with the first diagnosed episode of schizophrenia compared to healthy controls. However, no significant difference in PCS morphology was found in patients with first-episode affective psychosis [35]. 

Another brain area associated with hallucinations is the superior temporal sulcus. For example, a reduced depth of this sulcus is observed in patients with auditory hallucinations [36]. Some studies revealed that focal decreased gyrification might lead to hallucinations, but also global reduction of this process might play an important role [36,37]. Decreased cortical folding was also observed in patients with obsessive-compulsive disorder (OCD). Among these patients, the incidence of “prominent” or “present” PCS was significantly lower in the left hemisphere [38]. Altered connectivity of the PCG has also been correlated with generalized epilepsy [39] and resistance to some epileptic drugs [40].

Anatomical variations of gyral and sulcal patterns can probably be accompanied by variations of the cerebral arteries. As James et al. [41] stressed, anatomical variations of the circle of Willis are present in most subjects. This is also true regarding the anterior cerebral artery and its branches [42,43,44,45,46]. Knowledge of the arterial supply of the cingulate and paracingulate gyri may be clinically valid; for instance, this can help clinicians predict neuropsychological and neuroimaging features of isolated cingulate infarcts. Kumral et al. [47] suggested that the anterior and posterior cingulate complexes are functionally separate; the anterior cingulate complex plays a role in executive functions, episodic and working memory, set maintenance, while the posterior cingulate complex is focused on spatial and verbal attention, and the memory system. According to these authors, different patterns of cingulate infarcts are the result of variation in cingulate arterial supply or suggest a source of embolism [47]. Moreover, the pericallosal and callosomariginal arteries may be used during anterior cerebral artery bypass for complex aneurysms [48,49]. Acerbi et al. [49] described in situ side-to-side pericallosal–pericallosal artery and callosomarginal–callosomarginal artery bypasses for complex distal anterior cerebral artery aneurysms. Further studies are needed, however, to compare how anatomical variations of the cingulate and paracingulate gyri are related to arterial variations.


**• Study limitation**


The main limitation of the presented study is the lack of data on the medical history of the body donors. Moreover, the study was conducted in isolated hemispheres, so we could not compare consistency/inconsistency of the gyral pattern between right and left sides in the same brain. Comparing the degree of development of the PCG to the brain’s side on a population without neurological or psychiatric disorders should be performed. Further studies on the topic should also include information on handedness, sex, and the presence of various medical diseases. A critical question for future studies is whether any genetical/behavioral/environmental factors could have influenced some of the structural variations observed in this study. A valuable source of information would be studies regarding the connections of the PCG depending on the observed variant of the gyrus. Additionally, vascular variations should be examined in specimens with arteries injected with colored resin. The advantage of this work is that it draws attention to new perspectives in terms of anatomical and imaging studies of the cingulate and paracingulate gyri.

## 5. Conclusions

Simple anatomical classifications of brain cortex gyrification provide a valuable research model, although they may be insufficient in determining the actual functional areas of the cerebral cortex. Further efforts of gyral and sulcal pattern classifications should consider connections between a given gyrus and other areas of the brain. However, the awareness of the anatomical variability observed within the cortex is an essential starting point for in-depth research.

## Figures and Tables

**Figure 1 brainsci-11-00872-f001:**
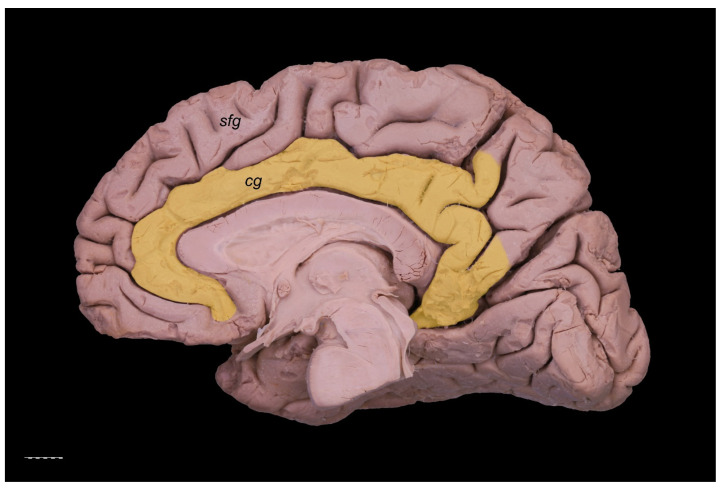
Absence of the PCG. The cingulate gyrus is marked pale yellow. Medial view of the right cerebral hemisphere. cg—cingulate gyrus; sfg—superior frontal gyrus. Scale bar corresponds to 10 mm.

**Figure 2 brainsci-11-00872-f002:**
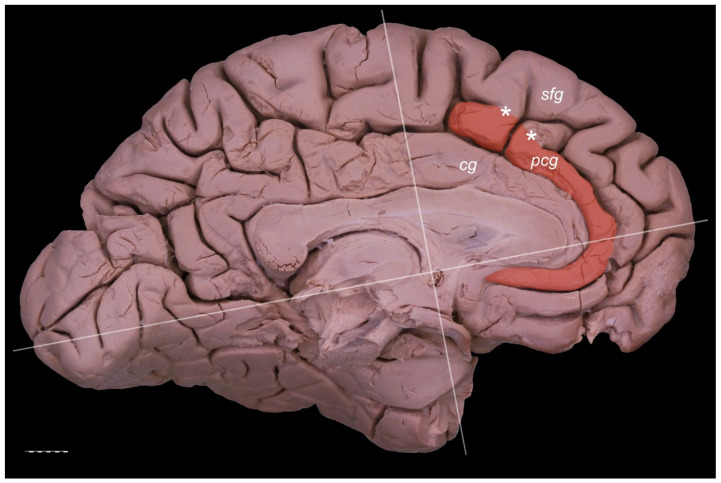
Presence of the PCG. Medial view of the left cerebral hemisphere. cg—cingulate gyrus; pcg—paracingulate gyrus; sfg—superior frontal gyrus. The posterior cingulate gyrus does not reach the level of the VAC line. Interruptions of the PCS (marked by white asterisks) are less than 20 mm. Scale bar corresponds to 10 mm.

**Figure 3 brainsci-11-00872-f003:**
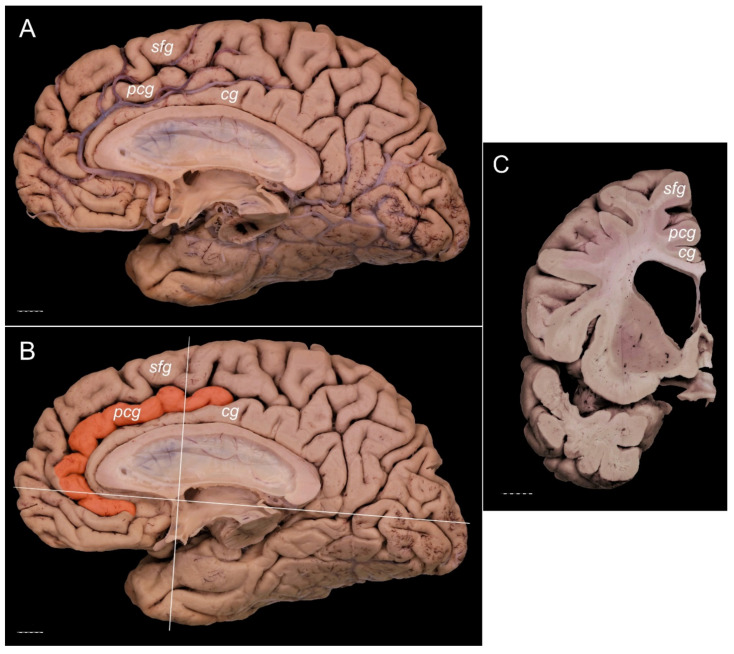
Presence of a prominent PCG. Specimens of the right cerebral hemisphere. The posterior cingulate gyrus crosses the level of the VAC line posteriorly. (**A**) General medial view of the hemisphere with callosomarginal artery running within the cingulate sulcus and sending branches that cross the PCG. (**B**) Medial view of the same specimen with the PCG colored orange. VAC and AC-PC lines are marked. (**C**) Coronal section of the specimen made 10 mm anterior to the VAC. Separate cingulate, paracingulate, and superior frontal gyri are well marked. cg—cingulate gyrus; pcg—paracingulate gyrus; sfg—superior frontal gyrus. The scale bar corresponds to 10 mm.

**Figure 4 brainsci-11-00872-f004:**
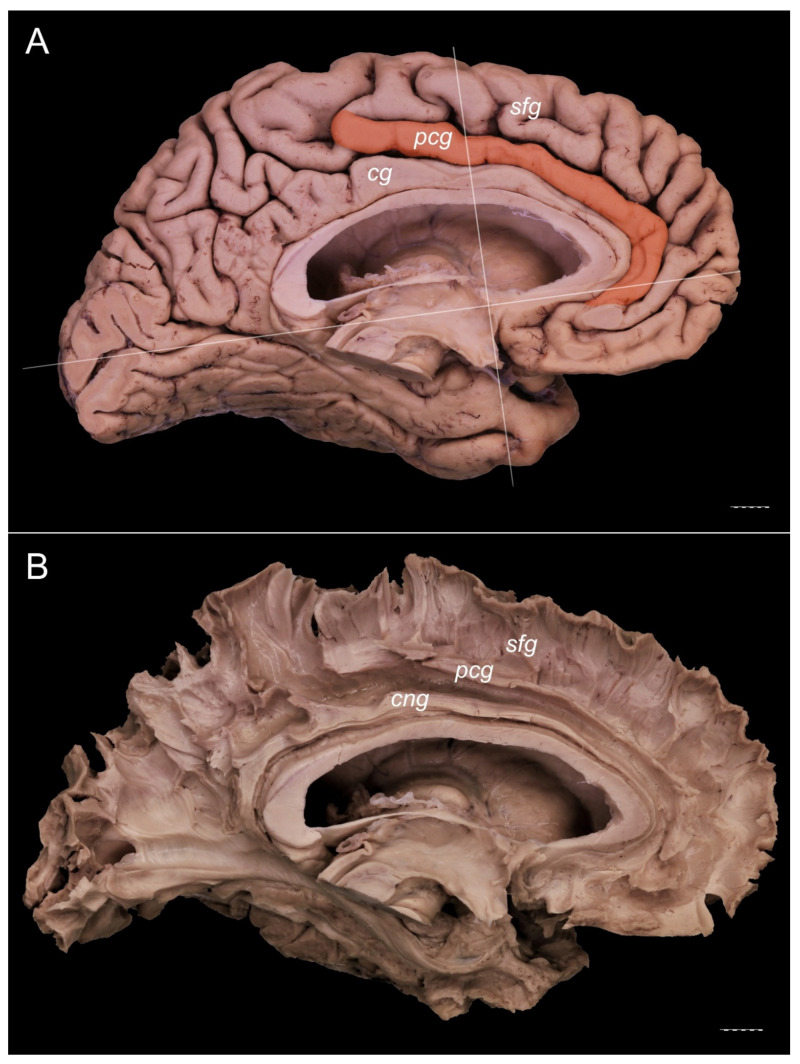
Presence of a prominent PCG. Medial view of the left cerebral hemisphere. The posterior cingulate gyrus crosses the level of the VAC line. No interruption of the gyrus is visible. (**A**) Specimen with a PCG (orange). The VAC and AC-PC lines are marked. (**B**) The specimen seen in Figure 4A, dissected using Klingler’s technique. A separate cingulum and white matter fibers of the paracingulate and superior frontal gyri are well developed. cg—cingulate gyrus; cng—cingulum; pcg—paracingulate gyrus; sfg—superior frontal gyrus. Scale bar corresponds to 10 mm.

**Figure 5 brainsci-11-00872-f005:**
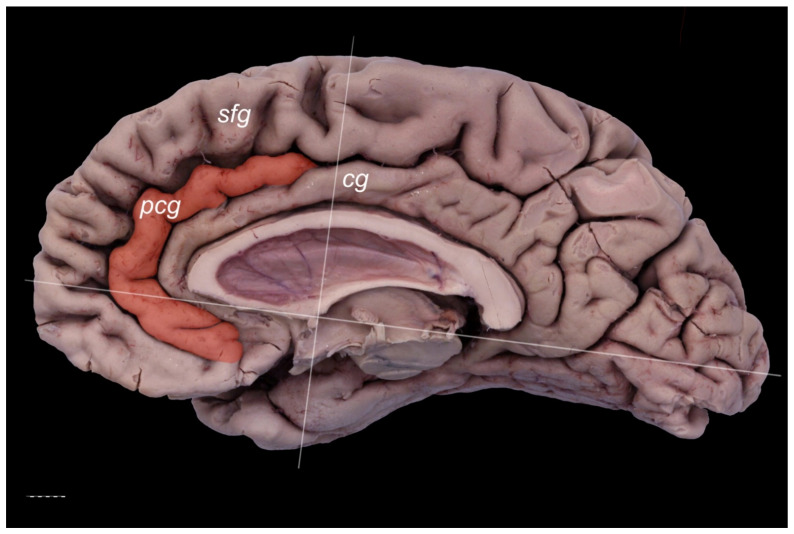
An ambiguous case of a PCG (orange). Medial view of the right cerebral hemisphere. The gyrus is well-developed, although only about 5 mm is missing in order to classify this specimen as “prominent” as it ends slightly anterior to the level of the VAC line. The VAC and AC-PC lines are marked. cg—cingulate gyrus; pcg—paracingulate gyrus; sfg—superior frontal gyrus. The scale bar corresponds to 10 mm.

**Figure 6 brainsci-11-00872-f006:**
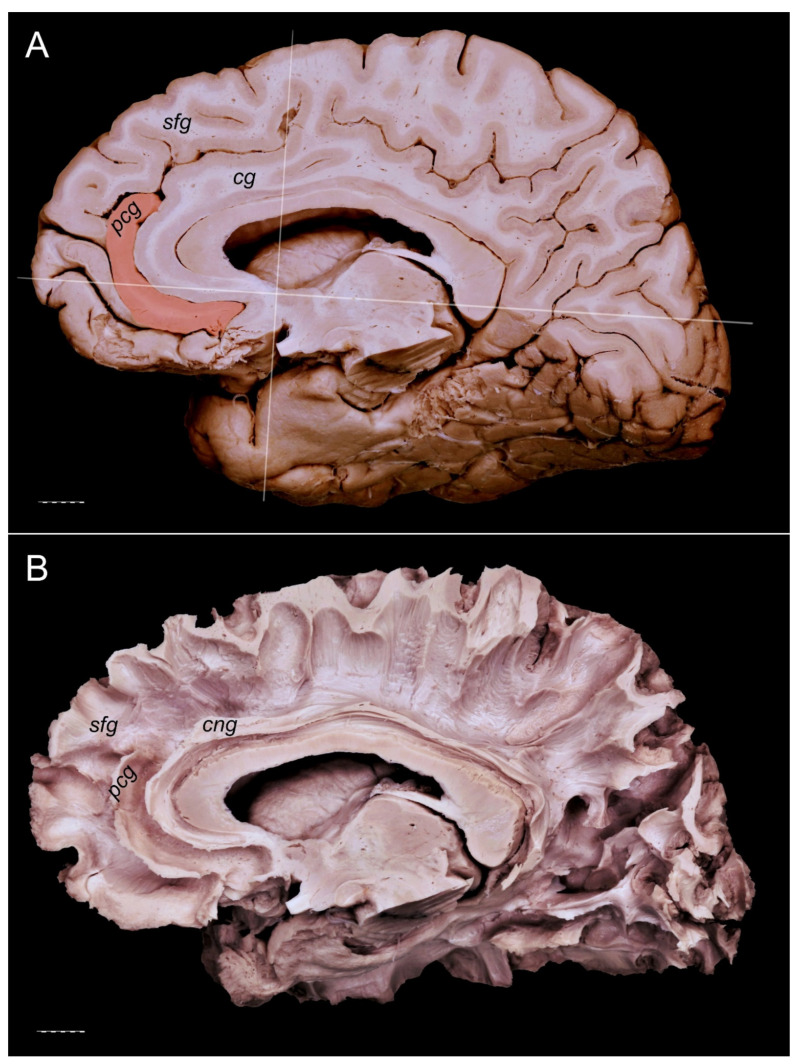
An ambiguous case. Medial view of the right cerebral hemisphere. This type can be classified as absent (PCS < 20 mm). From an anatomical point of view, based on a dissection performed using Klingler’s technique, it can also be classified as an “underdeveloped type.” (**A**) The vestigial PCG is colored orange. The VAC and AC-PC lines are marked. (**B**) The same specimen dissected using Klingler’s technique. The cingulum and white matter fibers of the superior frontal gyrus are visualized. Some white matter fibers of the PCG are exposed. cg—cingulate gyrus; cng—cingulum; pcg—paracingulate gyrus; sfg—superior frontal gyrus. The scale bar corresponds to 10 mm.

**Figure 7 brainsci-11-00872-f007:**
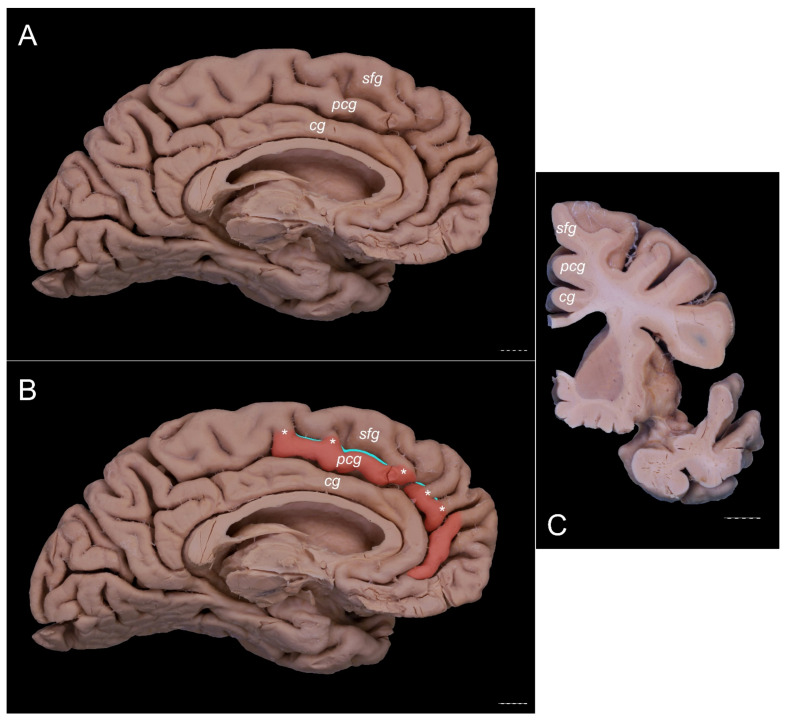
An ambiguous case. Specimens are of the left cerebral hemisphere. The posterior cingulate gyrus should be classified as present based on the length of the PCS < 40 mm. However, the gyrus crosses the level of the VAC line posteriorly. (**A**) A medial view of this specimen. (**B**) A medial view of the same specimen with the PCG colored orange. The VAC and AC-PC lines are marked. The interrupted PCS is marked with teal color. (**C**) Coronal section of this specimen at the level of the VAC line. A separate cingulate, paracingulate, and superior frontal gyri are marked. cg—cingulate gyrus; pcg—paracingulate gyrus; sfg—superior frontal gyrus. Scale bar corresponds to 10 mm.

**Figure 8 brainsci-11-00872-f008:**
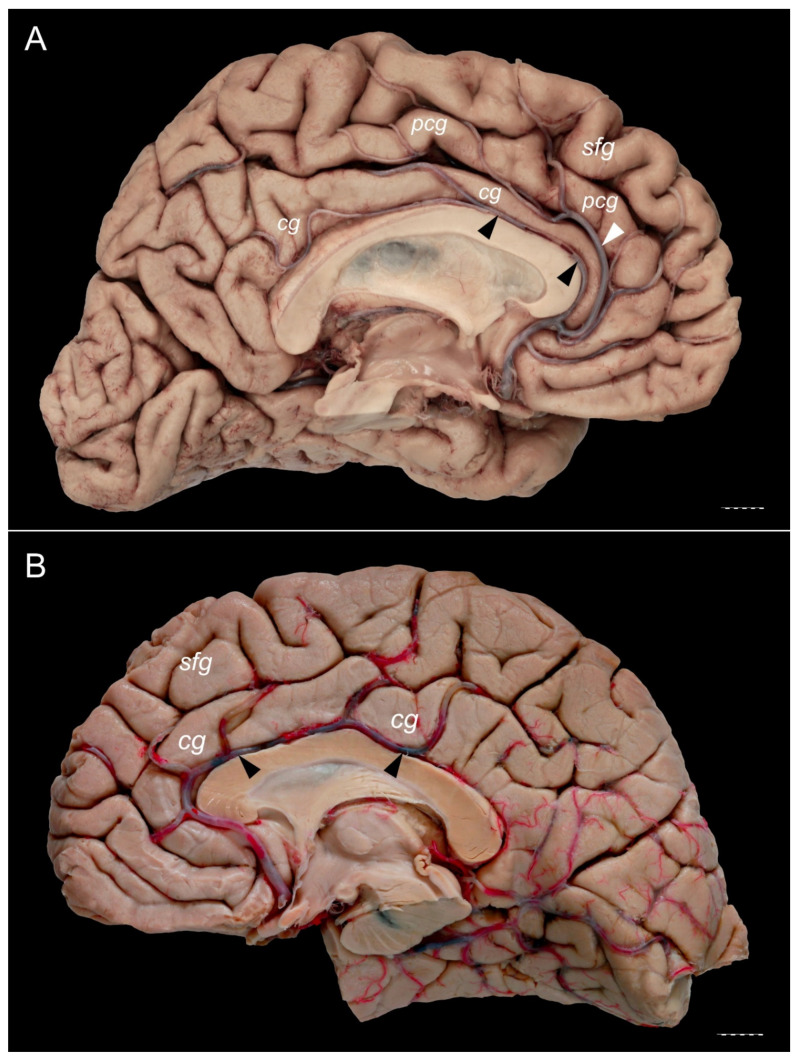
The relationships of the callosomarginal and pericallosal arteries. (**B**) Specimen showing a prominent type of the PCG. A pericallosal artery (marked by black arrowheads) and well developed callosomarginal artery (marked by white arrowhead) are present. Pericallosal artery is thinner than the callosomarginal artery on this specimen. (**A**) Specimen with absent PCG. Cingulate gyrus is well developed. Pericallosal artery (marked by black arrowheads) runs within the pericallosal sulcus, over the genu and superior surface of the body of the corpus callosum. cg—cingulate gyrus; pcg—paracingulate gyrus; sfg—superior frontal gyrus. Scale bar corresponds to 10 mm.

## Data Availability

Data are contained within the article.

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
