# Peer review of "Revisiting the Morphology and Classification of the Paracingulate Gyrus with Commentaries on Ambiguous Cases"

_brainsci, 2021, doi:10.3390/brainsci11070872_

Round 1

Reviewer 1 Report

The major weakness of the study is the use of isolated hemispheres without any clinical information. There is no information on handedness, gender, important medical diseases, etc. Unfortunately, there is no ability to match the hemispheres and look for compensatory changes.

The study is well done with careful documentation of the anatomical dissection. Use of the VAC and AC-PC line are appropriate. It would just be so much better to know what the other hemisphere looked like in these specimens. It would have been interesting as well to have specimens with intact vasculature to understand the relationships of the callosomarginal and pericallosal arteries.

Author Response

The major weakness of the study is the use of isolated hemispheres without any clinical information. There is no information on handedness, gender, important medical diseases, etc. Unfortunately, there is no ability to match the hemispheres and look for compensatory changes.

Answer: We agree with this crucial remark. This weakness of the presented study was stressed in the study limitations. However, the advantage of this work is that it draws attention to new perspectives in terms of anatomical and imaging studies of the cingulate and paracingulate gyri. We explained that further studies are needed regarding the connections of the PCG depending on the observed variant of the gyrus. The question also arises of how anatomical variations of the cingulate and paracingulate gyri are related to arterial variations. Thus, vascular variations of the callosomarginal and pericallosal arteries should be examined in specimens with arteries injected with colored resin.

The study is well done with careful documentation of the anatomical dissection. Use of the VAC and AC-PC line are appropriate.

Answer: Thank you for the kind words.

It would just be so much better to know what the other hemisphere looked like in these specimens. It would have been interesting as well to have specimens with intact vasculature to understand the relationships of the callosomarginal and pericallosal arteries.

Answer: We thank the reviewer for these suggestions. New Figure 8 presenting other specimens with intact vasculature was added to show the relationships and selected variations of the callosomarginal and pericallosal arteries.

Reviewer 2 Report

MTUI Comments: Revisiting the morphology and classification of the paracingulate gyrus with commentaries on ambiguous cases

Abstract:

I found the abstract to meet the appropriate length, includes a good summary and highlights original research findings.

Introduction:

It is clear and well organized. No changes needed.

Materials and Methods

  • Consider increasing the sample size in future studies.

  • Make an attempt to include past medical history

  • Include MRI studies in future studies

  • Description of the methodology is accurate, and the appropriate references have been cited

4.1 Limitations:

Future studies should include past medical history to validate the results with the various clinical syndromes resulting from the anatomical variations.

4.2 Conclusion:

The Anatomical outcomes confirmed other studies performed elsewhere as cited in the literature review.

A critical question for future studies is whether any genetical/behavioral/environmental factors could have influenced some of the structural variations observed in this study.

Bibliography:

Very well presented

Figures:

 Figure 1

Well presented

Figures 2, 3, 4 and 5

Well presented

Author Response

Response: We thank the reviewer for his thoughtful review of our work and kind words. We have thoroughly re-reviewed the manuscript and corrected any errors we came across. We thoroughly discussed the limitations of the study and took note of further research perspectives. We agree that further studies should be carried out based on imaging diagnostics on healthy individuals and people with various clinical syndromes. This work may serve as a starting point for such research.